## [Peer Review File · Nature Communications]

REVIEWER COMMENTS

Reviewer #1 (Remarks to the Author):

This manuscript demonstrates an approach to generate nonlinear relaxation (l -photon relaxation with $l \geq 1$), and thereby quantum Zeno dynamics and nonlinear Lamb shift, for a microwave superconducting resonator without the use of parametric drives. While there is prior art that covers some aspects of the work presented here, as I detail below, I argue that these results are indeed novel and worth publishing in Nature Communications.

The subject of this manuscript is timely, as bath engineering has become a central topic of research in the circuit QED community; for example with applications in bosonic quantum error correction, where for example cat code preparation relies on engineered two-photon dissipation. More generally in circuit QED, where microwave resonators are used to read out qubits, bath engineering might play a role in improving qubit measurement or coherence.

In this manuscript, bath engineering is done by galvanically coupling the resonator to a Josephson tunnel junction, whose bath is modeled by two electron reservoirs. Nonlinear relaxation can occur by an electron tunneling between the reservoirs across the junction, assisted by l photons from the microwave resonator. When energetically allowed, this process is not excluded by selection rules from the standard Hamiltonian governing the coupling of the tunneling electron to the superconducting phase difference across the junction, the manuscript's Eq. (1). However, such processes are non-negligible only if the resonator impedance Z_c becomes comparable to the quantum of resistance R_Q , equivalently by arranging that phase fluctuations be comparable to charge fluctuations [cf. e.g. Masluk et al. PRL 109, 137002 (2012)],

This work achieves a sufficiently large $Z_c \sim 4.5$ kOhm ($R_Q \sim 6.5$ kOhm) by the use of granular aluminum for the resonator. In this regime, bias voltage across the junction can be set so that dissipation is only sizable for $l \geq 2$, which allows the authors to exemplify quantum Zeno dynamics in the lowest manifold consisting of the states $|0\rangle$ and $|1\rangle$. The highly nonlinear dynamics is excellently captured by master equation simulations detailed in the SI.

Regarding connections to previous work, understanding of the theory of voltage tuned reservoir engineering has largely been developed in:

Catelani et al. in Phys. Rev. B 84, 064517 (2011) (<https://arxiv.org/abs/1106.0829>) for the quasiparticle-induced relaxation channels of a superconducting qubit.

Silveri et al. Phys. Rev. B 96, 094524 (2017) (<https://arxiv.org/abs/1706.07188>).

In particular, Silveri et al. 1706.07188 proposed a similar setup, but essentially different in that it used low-impedance coplanar waveguide resonators, and therefore the matrix elements responsible for multi-photon transition would be highly suppressed. However, that work already mentions that: "Despite of this suppression of the matrix elements, we note that photon-assisted tunneling drives multi-photon transitions, formally, without typical selection rules. In certain conditions, due to the enhancement by electron tunneling rates, the rate of absorptive two-photon transitions can exceed the emissive single-photon rate (...). In this paper, however, we do not concentrate on the details of the high-photon-number processes." The more recent preprint Esteve et al. (<https://arxiv.org/abs/1807.02364>), upon which the present manuscript builds, covered the master equation in the high-impedance regime.

Experiments with granular aluminum resonators reaching high impedance have been performed:

Gruenhaupt et al. Phys. Rev. Lett. 121 117001 (2018), and Maleeva et al. Nat. Comm. 9, 3889 (2018). An experiment with a similar setup, though not high-impedance and using a NIS junction, were published by Silveri et al. in Nat. Phys. 15, 533 (2019). Since that experiment does not reach the high impedance regime, quantum Zeno dynamics was inaccessible.

Essentially this work exemplarily puts together the two ingredients mentioned above: a high-impedance linear mode implemented with granular aluminum; and voltage-tuned reservoir engineering by coupling to a Josephson tunnel junction. I believe that both the quality of the experimental data in this new regime exhibiting quantum Zeno dynamics (saturation and power broadening similar to a two-level system, nonlinear Lamb shift etc.), and the excellent agreement with the theory (e.g. Figures 3, 4), make this paper worth publication in Nature Communications.

The paper is very clearly written with great detail on the experimental and very fine theoretical analysis provided in the SI. I therefore only have one minor point: in the regime where the oscillator has effectively turned nonlinear, it would be worthwhile to quote the resulting anharmonicity (regarding Figure 4). Can the authors also quote the incident power for Fig. 4 (e.g. in the sentence "The power is sufficiently low..."), in analogy to their having provided the power for Fig 2b?

Reviewer #2 (Remarks to the Author):

The authors implement multi-photon loss in a microwave resonator. The working principle is photon assisted quasi-particle tunneling in a Josephson junction. The resonator has very high impedance,

close to the superconducting resistance quantum which results in matrix elements for n -photon loss that depend only weakly on n . By choosing a bias voltage of the junction where at least two photons are required to provide the energy for a Cooper pair to tunnel, they reach a regime where 2-photon loss is significantly stronger than 1 photon loss. Their data indicates that the resonator state is effectively limited to the $|0\rangle |1\rangle$ subspace, i.e. a two level system.

While the key ingredients for this work, i.e. the physics of photon assisted tunneling, the fact that high impedances increase nonlinearity of inelastic charge tunneling, and the design of high-impedance resonators are well established, the experimental realization of dominant multi-photon loss is, to my knowledge, original and new. I doubt, however, that the implemented two-photon loss is sufficiently pure for e.g. the protocols of ref 8. Given the rather high characteristic impedance of the mode I would expect that only energetically forbidden processes (i.e. 1 photon loss) are suppressed, but that processes involving more than 2 photons also contribute. Also, the resonator seems to have low internal quality factor ($\kappa = 4.2$ MHz), i.e. one-photon loss. Is this loss understood or could it be improved?

In my opinion, the strength of the paper lies more in the very detailed understanding of the observed absorption rates and photon-number-dependent frequency shifts rather than in the implementation of two photon loss. The authors indeed achieve an impressive agreement between their theoretical models and experimental data. All analysis is solidly supported by the very complete supplementary material. This information makes it possible to reproduce the results and the theoretical modeling, especially of the Lamb shift, will certainly be of use to many groups in the field. Beyond the field, I would expect the paper to have a rather limited impact.

I think the paper will mostly appeal to a specialist audience (even in case the engineered bath with dominant two-photon is practical) and should be published in a more specialized journal (in a form close to the one presented here).

Before publication, the following more detailed remarks should be addressed:

1. A restriction to the $0, 1$ subspace of a resonator has been proposed and achieved in a similar system based on inelastic Cooper pair tunneling (Phys Scr. T165 014029, PRA 93 060301, ref 27 (which is only mentioned in the context of implementing high impedances)). The mechanism presented there is also based on the high impedance of the resonator but not entirely the same (suppression of the $\langle 2|$ to $|1\rangle$ matrix element). PRA 97 013855 also proposes an implementation of n -photon loss based on inelastic Cooper pair tunneling with in a junction coupled to a high-impedance mode. These related works should be mentioned.

The resonator impedance the authors have implemented seems indeed better suited for the approach of Phys Scr T165 and ref 27 than the device in ref 27. Have the authors tried this?

2. In this context, because of the high characteristic impedance of the higher modes of the resonator, I would have expected these modes to be significantly populated due to inelastic Cooper pair tunneling at specific bias voltages. I would then expect these populated modes to affect the observed mode (e.g. via ac Stark shift). At what bias voltages are these resonances expected and did the authors observe any signatures?

3. The authors mention that they have induced Zeno dynamics but this is not substantiated (at least in the main text). Under Zeno dynamics I understand that frequent protective measurements restrict the system to a subspace due to the quadratic dependence of probability on amplitude, but the main text sounds like any restriction to a subspace would do. Zeno dynamics may play a role here, but because the measurement in this case is destructive --it puts the resonator in the 0 state, inside the restricted subspace-- it is not clear if the system stays in the 0,1 space or enters the 2 and decays to the 0 state. In order to show that the restricted dynamics the authors observe is indeed Zeno dynamics they would have to show that the system never actually reaches the 2 state. i.e. there is no PAT and no DC current flowing through the junction. Could the authors use Fig 8 in the SI to prove the Zeno dynamics?

4. Have the authors quantified the intrinsic nonlinearity of the granular AI resonator? Does it play a role at the photon numbers at play here?

5. The authors should use consistent units for pump power. In figure 2 they use dBm which I know, in figure 3 they use η in MHz which is only defined in the SI, in figure 4 it is "sufficiently low to mostly probe the 0 \rightarrow 1 transition", which is a very fuzzy unit.

6. Line 154: "to the one the" \rightarrow "to the"

7. SI page 2: Table I:

the characteristic impedances of modes 7 to 16 seem to be in Ohm not in kOhm as indicated.

Caption: Z_n instead of Z_c

8. SI page 7: the approximation to only keep $I_n = 0$ seems not well justified given that many modes have high characteristic impedance and matrix elements transferring photon energy between modes should be non negligible.

SI Page 8: "A plot of $|\langle a \rangle|^2$ at resonance is shown in figure 2a". This should be figure 3a.

We thank both referees for their careful reading of the manuscript. We are grateful to referee #1 for his opinion on our work. In view of the criticisms of referee #2, we have expanded the manuscript to clarify why our results are of interest beyond the circuit QED community. The revised manuscript now includes a discussion about the photoelectric effect and why our results are incompatible with the usual semiclassical treatment of this effect.

Also the referee compares a lot our work to previous work on inelastic Cooper pair tunneling. We stress that inelastic Cooper pair tunneling is a coherent process that appears as a non-linear drive in the system Hamiltonian. Quasiparticle tunneling is fundamentally different, it is intrinsically non coherent and can only be described through a master equation approach. Therefore, the two effects fall in different categories. While quasiparticle tunneling can be considered as a quantum bath engineering technique, we believe that inelastic Copper pair tunneling may not.

We now turn to a detailed discussion of the referees' comments:

Referee #1:

- In the regime where the oscillator has effectively turned nonlinear, it would be worthwhile to quote the resulting anharmonicity.

We added the resulting anharmonicity in the main text (line 213).

- Can the authors also quote the incident power for Fig. 4 (e.g. in the sentence "The power is sufficiently low..."), in analogy to their having provided the power for Fig 2b?

We now explicitly give the power used for this measurement in the caption of figure 4.

Referee #2:

- I doubt, however, that the implemented two-photon loss is sufficiently pure for e.g. the protocols of ref 8. Given the rather high characteristic impedance of the mode I would expect that only energetically forbidden processes (i.e. 1 photon loss) are suppressed, but that processes involving more than 2 photons also contribute.

We agree with the referee that processes, which correspond to the loss of an odd number of photons, break the parity symmetry that is used in the protocol of ref 8. We never pretend in the manuscript that our bath engineering technique could be readily applied to cat state stabilization. We have moved the sentence citing ref 8 to avoid any confusion (line 24).

- Also, the resonator seems to have low internal quality factor ($\kappa = 4.2$ MHz), i.e. one-photon loss. Is this loss understood or could it be improved?

We agree with the referee that the remnant single photon loss, or more precisely the ratio of two to one photon loss as given in the manuscript is the key figure of merit for our bath engineering technique. In theory, this ratio, presently of 25, could be much higher. There is no clear theoretical bound to its value, except the one due to the intrinsic quality factor of the resonator. In the present sample, the lowest observed intrinsic loss is around 0.5 MHz, which happens at lower voltage. Resonator test samples, fabricated together with the one examined in this paper, but not connected to a junction, exhibit an intrinsic loss rate of 0.2 MHz. Other groups have managed to obtain a linewidth of 70 kHz in GrAl resonators (see e.g. ref 24). This leaves plenty of room for improvement. We believe that the observed remnant single photon loss may be due to quasiparticle poisoning. The out of equilibrium quasiparticles may tunnel and absorb a photon, even when the voltage difference forbids this process for quasiparticles at equilibrium. The sample was not particularly designed to minimize this effect (there is no quasiparticle trap), also a better shielding of the sample may reduce this effect. However, we do not have a direct proof that this poisoning is the source of the single photon loss.

- A restriction to the 0, 1 subspace of a resonator has been proposed and achieved in a similar system based on inelastic Cooper pair tunneling (Phys Scr. T165 014029, PRA 93 060301, ref 27 (which is only mentioned in the context of implementing high impedances)). The mechanism presented there is also based on the high impedance of the resonator but not entirely the same (suppression of the $\langle 2|$ to $|1\rangle$ matrix element).

We are well aware work of these works on Inelastic Cooper Pair Tunneling (ICPT). As mentioned above, we believe that ICPT is fundamentally different from quasiparticle (QP) tunneling. ICPT introduces a non-linear drive in the Hamiltonian. In conjunction with standard dissipation (single photon loss) and by using adiabatic elimination, ICPT may be used for state stabilization as proposed, for example, in PRA 93 060301. This approach is then similar to the usual bath engineering in circuit QED except for the fact that the coherent drive of a non-linear oscillator is replaced by ICPT. This approach requires a very fine tuning of the characteristic impedance in order to cancel the unwanted matrix element, which is difficult to realize experimentally. To our knowledge, engineered dissipation with ICPT, as proposed in PRA 93 060301 (or in any other form), has not yet been realized experimentally. So far, it has been demonstrated that ICPT can be used as a non classical source of microwave photons, which is conceptually very different from engineered dissipation.

- PRA 97 013855 also proposes an implementation of n-photon loss based on inelastic Cooper pair tunneling with in a junction coupled to a high-impedance mode. These related works should be mentioned.

To our understanding, the reference PRA 97 013855 proposes to convert a single photon to n photons using ICPT as a parametric amplification mechanism. It is not equivalent to a n photon loss process. Our paper being focused on QP tunneling, we have chosen not to cite all papers from the abundant literature about ICPT.

- In this context, because of the high characteristic impedance of the higher modes of the resonator, I would have expected these modes to be significantly populated due to inelastic Cooper pair tunneling at specific bias voltages. I would then expect these populated modes to affect the observed mode (e.g. via ac Stark shift). At what bias voltages are these resonances expected and did the authors observe any signatures?

Yes, we do observe a lot of ICPT in our sample. These results will be published in a separate paper. We have added a sentence to clarify this point (line 139-141). Once again, because ICPT is very different from QP tunneling and because ICPT has already been extensively studied, we have decided to focus the paper on the QP tunneling, which we believe is more original. ICPT appears at voltages such that $2eV=hf$, where f is a combination of the resonant frequencies present in the system ($f=n_1 f_1 + n_2 f_2 + \dots$). At the voltages considered in this paper, only high frequency modes contribute, which are weakly coupled to the junction. Therefore, ICPT has very little influence in the considered voltage range.

- The authors mention that they have induced Zeno dynamics but this is not substantiated (at least in the main text). Under Zeno dynamics I understand that frequent protective measurements restrict the system to a subspace due to the quadratic dependence of probability on amplitude, but the main text sounds like any restriction to a subspace would do. Zeno dynamics may play a role here, but because the measurement in this case is destructive --it puts the resonator in the 0 state, inside the restricted subspace-- it is not clear if the system stays in the 0,1 space or enters the 2 and decays to the 0 state. In order to show that the restricted dynamics the authors observe is indeed Zeno dynamics they would have to show that the system never actually reaches the 2 state. i.e. there is no PAT and no DC current flowing through the junction. Could the authors use Fig 8 in the SI to prove the Zeno dynamics?

We agree with the referee that the measurement induced by the junction is only non-destructive in the $\{|0\rangle, |1\rangle\}$ subspace and not in the orthogonal subspace. Therefore Quantum Zeno Dynamics (QZD) is only induced in the $\{|0\rangle, |1\rangle\}$ subspace and not in the orthogonal one. We have modified the main text to clarify this point (line 151-154). If the initial state lies in the $\{|0\rangle, |1\rangle\}$ subspace, the high loss from states with more than two photons prevents them from being populated, without photons being lost. This realizes a QND measurement: the absence of tunneling measures the state being in the $\{|0\rangle, |1\rangle\}$ subspace. As long as there is no quantum jump corresponding to one tunneling event, there is no difference if QZD is induced by a strong loss or by a two orthogonal QND measurements, both lead to the exact same quantum trajectories in the $\{|0\rangle, |1\rangle\}$ subspace. We agree with the referee that when a quantum jump occurs, signaling that the state has left the $\{|0\rangle, |1\rangle\}$ subspace, the following evolution is different in the case of orthogonal QND measurements and in our case. In the first case, the system then evolves in the $\{|2\rangle, |3\rangle, \dots\}$ subspace, while, in our case, the state is (most of the time) reset to $|0\rangle$. As suggested by the referee, we now include the results of a simulation of QZD generated by orthogonal QND measurements in figure S9. In the region where the dynamic is restricted to the $\{|0\rangle, |1\rangle\}$ subspace, we find that both models are equivalent as expected.

- Have the authors quantified the intrinsic nonlinearity of the granular AI resonator? Does it play a role at the photon numbers at play here?

We now estimate the non-linearity in the SI. It is about 200 Hz per photon, which is negligible.

- The authors should use consistent units for pump power. In figure 2 they use dBm which I know, in figure 3 they use η in MHz which is only defined in the SI, in figure 4 it is "sufficiently low to mostly probe the 0 \rightarrow 1 transition", which is a very fuzzy unit.

We now include the definition of η in the main text (line 151) and clarify the value of the pump used in figure 4. We introduce η when we describe the QZD because it sets the scale to which the projective measurement rate should be compared to.

- SI page 2: Table I: the characteristic impedances of modes 7 to 16 seem to be in Ohm not in kOhm as indicated. Caption: Z_n instead of Z_c

We have corrected the values, which were indeed wrong, and modified the caption.

-SI page 7: the approximation to only keep $I_n = 0$ seems not well justified given that many modes have high characteristic impedance and matrix elements transferring photon energy between modes should be non negligible.

Figure S6 precisely quantifies this effect. The dashed line corresponds to the approximation where we only keep $I_n = 0$ terms. The solid line takes into account multimode processes. Since the multimode processes do not significantly change the main features of the one and two photon loss terms, we have decided to not mention them in the main text.

- SI Page 8: "A plot of $||^2$ at resonance is shown in figure 2a". This should be figure 3a.

We have corrected the sentence.

REVIEWERS' COMMENTS

Reviewer #1 (Remarks to the Author):

After carefully studying the Authors' answers and revisions, especially those to the other Referee, I conclude that the authors have very satisfactorily addressed all concerns. Notably, the new version of the manuscript contains a detailed analysis of the quantum Zeno dynamics; and a restatement of the problem in terms of a tunneling analogue of the photoelectric effect, while proving that the regime reached here eludes a semiclassical description.

Moreover, with regards to the objection that the current protocol may not be useful for quantum engineering applications in bosonic error correction, I agree that the purpose of this paper was not to provide an alternative route to engineer two-photon dissipation *only*. This proof of principle for a special type of bath engineering of a high-impedance mode, based on quasiparticle tunneling, is valuable in its own right.

As I have emphasized in my first review, this work brings together two ideas that were found in the literature before, that of achieving a high-impedance linear mode due to use of granular aluminum; and that of reservoir engineering by voltage tuning an SIS Josephson junction coupled to the resonator. The quality of the experimental data in this new regime and the excellent agreement with a theory that is quite involved make this paper worth publication in Nature Communications.

Reviewer #2 (Remarks to the Author):

The authors have removed most of my concerns from the previously revision and I think the paper is ready for publication after a few minor corrections (see below). I still believe the paper will mostly interest a specialist audience but the findings are certainly important and new, and after comparing with other recent publications in Nature Communications, I think the paper fits here.

I would still argue that inelastic quasi particle (IQPT) and Cooper pair tunneling (ICPT) are not fundamentally different from the perspective of bath engineering because the nonlinearity used to generate a non-thermal synthetic bath from a thermal bath is the same (within a factor of 2^2). The only difference being that there is an intermediate thermal bath composed of essentially quasi-particle expectations for quasi-particle tunneling or electromagnetic modes for Cooper pair

tunneling, but in the end both thermalize with phonons. However, bath engineering with ICPT has not been realized experimentally either, and I can definitely understand the line the authors have drawn for including references.

The discussion about Zeno Dynamics is now much clearer and figure S9 very helpful. However, figure S9 seems to indicate that above $\eta \approx 2\pi \times 3$ MHz Zeno dynamics is not explaining the restriction to the $\{|0\rangle, |1\rangle\}$ subspace which continues to exist up to $2\pi \times 10$ MHz. This seems to indicate that dissipation becomes important, which is also visible in figure S8, where the loss rate in excess of intrinsic loss starts to become comparable to η at around $2\pi \times 3$ MHz as well.

I would therefore conclude that there is a transition from Zeno dynamics to a dissipative restriction to the $\{|0\rangle, |1\rangle\}$ with different dynamics before the restriction breaks down completely at around $\eta \approx 2\pi \times 10$ MHz. This should be stated more clearly in the main text.

figure 4: The pump power should (also) be given in terms of η to be able to compare with figure 3.

line 209: Fig 1a is the experimental setup. I do not think this is the right reference here.

line 213: Why do you quote the nonlinearity for a bias where you have not measured it?